# Physics-Informed GNN for Non-linear constrained optimization: PINCO a solver for the AC-Optimal Power Flow

## Abstract

The energy transition is driving the integration of large shares of intermittent power sources in the electric power grid. Therefore, addressing the AC optimal power flow (AC-OPF) effectively becomes increasingly essential. The AC-OPF, which is a fundamental optimization problem in power systems, must be solved more frequently to ensure the safe and cost-effective operation of power systems. Due to its non-linear nature, AC-OPF is often solved in its linearized form, despite inherent inaccuracies. Non-linear solvers, such as the interior point method, are typically employed to solve the full OPF problem. However, these iterative methods may not converge for large systems and do not guarantee global optimality. This work explores a physics-informed graph neural network, PINCO, to solve the AC-OPF. We demonstrate that this method provides accurate solutions in a fraction of the computational time when compared to the established non-linear programming solvers. Remarkably, PINCO generalizes effectively across a diverse set of loading conditions in the power system. We show that our method can solve the AC-OPF without violating inequality constraints. Furthermore, it can function both as a solver and as a hybrid universal function approximator. Moreover, the approach can be easily adapted to different power systems with minimal adjustments to the hyperparameters, including systems with multiple generators at each bus. Overall, this work demonstrates an advancement in the field of power system optimization to tackle the challenges of the energy transition. The code and data utilized in this paper are available at https://anonymous.4open.science/r/opf_pinn_iclr-B83E/.

## 1 Introduction

Power systems aim to ensure a reliable delivery of electricity to consumers. In a steady-state operation, power plants generate electric energy to meet the load demand while maintaining the voltage and frequency across the network within the required limits. The steady-state characteristic of the power system is described by the non-linear sinusoidal power flow equations, commonly known as the power flow (PF) problem. Additionally, by allocating electricity generation in a cost-effective manner throughout the day, the optimal power flow (OPF) problem is solved. This involves finding the most economical generation dispatch that satisfies the power flow equations and operational constraints. When no simplifications are made to the sinusoidal nature of the power flow equations, the problem is referred to as the alternating current optimal power flow problem (AC-OPF). The problem is used both in operation, being solved up to once every 5 minutes Nair et al. (2024) and planning.

Given the nonlinearity of the power flow equation, this problem is inherently non-convex and is NP-hard Pineda et al. (2023). Traditionally, different approaches are used to solve the OPF. The easiest method is using a linear form of the power flow equation, called DC approximation, based on small-angle approximations. Using the DC power flow model brings well-known limitations Kile et al. (2014); Baker (2020), leading to sub-optimal solutions of the complete AC-OPF. Alternatively, convex relaxation techniques have been extensively examined Lavaei & Low (2010); Low (2014); Molzahn & Hiskens (2016). Currently, nonlinear programming methods such as the interior point method are established approaches to solving the OPF Zimmerman & Wang (2016). Nevertheless,

these methods still face significant computational burdens and scalability challenges when applied to large-scale systems, limiting their utilization.

The limitations of traditional methods have driven researchers to explore alternative solutions using machine learning techniques. One of the first approaches to solving the power flow problem is presented by Donon et al. (2020). They introduce a graph neural network (GNN) solver for the AC power flow problem, employing a supervised learning approach. Their pioneering work utilized GNNs, a deep learning architecture beneficial for non-Euclidean data structures, like power grid datasets. Recent work Piloto et al. (2024) exploring the use of Graph Neural Networks (GNNs) for power systems, particularly for optimal power flow (OPF), demonstrates that GNNs can be effectively trained in a supervised manner to address the computationally intensive security-constrained AC-OPF problem Capitanescu et al. (2011). These studies utilize supervised learning techniques to establish a mapping between loading conditions and AC-OPF solutions. However, a computational burden due to using a conventional solver to create the dataset is always involved. Furthermore, the conventional solver does not necessarily find the global optimum. With the advent of physics-informed neural networks (PINNs) Raissi et al. (2019) and the subsequent growth of scientific machine learning, approaches that extend beyond pure data-driven learning have emerged. PINNs leverage physical laws to guide the training of neural networks, minimizing the need for labeled data. A comprehensive review of PINN applications in the power system domain is provided in Huang & Wang (2023). In Nellikkath & Chatzivasileiadis (2022), the authors presented a deep neural network trained using both data and physical equations, specifically incorporating the AC-OPF KKT conditions. The works of Huang et al. (2024) and Chen et al. (2022) propose deep neural networks for solving the AC-OPF without requiring labeled data, while also utilizing traditional methods to solve the nonlinear power flow equations. However, these approaches employ deep neural networks that lack the ability to generalize across different power grid topologies. Notably, Owerko et al. (2022) combined GNNs with physics-informed learning to solve the AC-OPF in an unsupervised manner. This approach enables solutions to the non-convex AC-OPF without relying on other solvers, thus avoiding bias. Nevertheless, Owerko et al. (2022) did not obtain OPF solutions without violating constraints, and they did not test their method on power systems with more than one generator per electrical bus.

To our knowledge, no existing method employs physics-informed neural networks (PINNs) in a fully end-to-end unsupervised manner without constraint violations that can compete with traditional solvers for solving the AC-OPF. Furthermore, we observe that previous works do not test the capability of such solvers in power systems where multiple generators per bus are present.

In this work, we tackle these research gaps, proposing PINCO. We developed a novel approach for the AC-OPF that combines GNNs and a variation of the conventional PINN paradigm that accounts for problems with hard constraints, named H-PINN Lu et al. (2021). PINCO allows for solving the AC-OPF without violations, and using GNN allows for easy adaptation to different power grid topologies and scalability. Furthermore, we provide a modeling approach dealing with multiple electricity generators per bus. Finally, we evaluate our developed methodology's computational efficiency and performance compared to established non-linear optimization solvers, demonstrating that this approach can compete with state-of-the-art solutions. In this work, we assess PINCO's potential as a nonlinear programming solver and test the canonical ability of neural networks as universal function approximators. We argue that the primary advantage of our approach lies in its speed in providing feasible solutions and its ability to generalize to various inputs and power system topologies. We test our approach on the IEEE9, IEEE24, IEEE30, and IEEE118 bus systems, covering a wide range of power grid topologies.

In this work, we contribute to the research community as follows:

- We introduce an unsupervised physics-informed GNN architecture capable of solving the AC-OPF without inequality constraint violations.

- We prove its ability to act as a solver on a single loading condition and as a universal function approximator that can act as a solver on unseen loading conditions.

- We test a method that leverages scientific machine learning to solve constrained non-linear optimization problems.

- We show that our approach can be adapted to different power systems with minimal changes to the hyper-parameters, including power systems with multiple generators per bus.

The paper is structured as follows: Section 2 outlines the methodology; Section 3 introduces the experimental setting and evaluation metrics; Section 4 presents the results on benchmark IEEE bus systems; Section 5 presents the limitations of this study; Section 6 concludes with final remarks and discussion.

## 2 METHODOLOGY

In this section, we provide a detailed definition of the AC-OPF and introduce the two core components of our PINCO method: GNN and physics-informed neural networks with hard constraints.

### 2.1 THE OPTIMAL POWER FLOW PROBLEM

The Optimal Power Flow (OPF) problem is a crucial optimization challenge extensively applied in modern energy systems. It involves a network of electrical buses (i.e., connection points between power lines, where load or generators can be located), denoted by $N$, interconnected by $E$ branches, (i.e., power lines or transformers). Each bus may host one or more generators, which inject power into the system, and loads, which consume it. The objective of the OPF problem is to minimize the total generation cost while adhering to the system's physical constraints. In this work, we address the full alternating current (AC) version of this model, which more accurately represents real-world grid conditions. Specifically, we consider a set of generator buses $N_g$ and load buses $N_d$. The AC-OPF problem can be formulated as follows, where $C_r$ represents the cost associated with operating generator $r$:

$$
\begin{aligned}
\min_{P_{g,i}, Q_{g,i}, V_i, \theta_i} \quad & \sum_{r \in N_g} C_r(P_{g,r}) \quad \forall r \in N_g \\
\text{s.t.} \quad & h_i(P_{g,i}, Q_{g,i}, V_i, \theta_i, P_d, Q_d) = 0 \quad \forall h_j \in H \\
& g_j(P_{g,i}, Q_{g,i}, V_i, \theta_i, P_d, Q_d) \le 0 \quad \forall g_l \in G
\end{aligned}
\tag{1}
$$

The equality constraints $H$ represent the nodal balance equations:

$$
H = \bigcup_{i \in N} \left\{ P_{g,i} - P_{d,i} - g_i^{sh} = \sum_{(ij) \in E} p_{ij} \right\} \cup \left\{ Q_{g,i} - Q_{d,i} + b_i^{sh} = \sum_{(ij) \in E} q_{ij} \right\}
\tag{2}
$$

Here, the active power demand and generation at bus $i$ are denoted by $P_{d,i}$ and $P_{g,i}$ (in MW), while the reactive power demand and generation are $Q_{d,i}$ and $Q_{g,i}$ (in MVAr). The active power $p_{ij}$ and reactive power $q_{ij}$ flowing between buses $i$ and $j$ are governed by the power flow equations:

$$
\begin{aligned}
p_{ij} &= g_{ij}(\tau_{ij} V_i^2) - V_i V_j \left( b_{ij} \sin(\theta_{ij}) + g_{ij} \cos(\theta_{ij}) \right) \\
q_{ij} &= -(g_{ij} + \frac{Sh_{ij}}{2})(\tau_{ij} V_i^2) - V_i V_j \left( g_{ij} \sin(\theta_{ij}) - b_{ij} \cos(\theta_{ij}) \right)
\end{aligned}
\tag{3}
$$

Here, $V_i$ is the voltage magnitude at bus $i$ (in volts), and $\theta_i$ is the phase angle (in degrees), with the phase angle difference between buses $i$ and $j$ denoted by $\theta_{ij}$. The grid characteristics include the conductance $g_{ij}$ and susceptance $b_{ij}$ of the transmission lines, shunt admittances $Sh_{ij}$[1], and transformer tap ratios $\tau_{ij}$. In addition, shunt elements, such as capacitors or inductors, are represented by fixed admittances to the ground: $g_i^{sh}$ (MW consumed) and $b_i^{sh}$ (MVAr injected).

The inequality constraints on the generator limits, for active and reactive power at each bus, are defined as:

$$
G_P = \bigcup_{r \in N_g} \{ P_{G,\min} \le P_g \le P_{G,\max} \}, \quad G_Q = \bigcup_{r \in N_g} \{ Q_{G,\min} \le Q_g \le Q_{G,\max} \}
$$

---

[1]Shunt admittance represents the admittance between a bus and ground in the power system.

Similarly, the inequality constraints on voltage magnitudes $G_V$ and branch thermal limits $G_S$ are:

$$G_V = \bigcup_{i \in N} \left\{ V_{i,\min} \leq V_i \leq V_{i,\max} \right\}, \quad G_S = \bigcup_{(ij) \in E} \left\{ (p_{ij})^2 + (q_{ij})^2 \leq |S_{\max,ij}| \right\}$$

where $|S_{\max,ij}|$ represents the maximum allowable apparent power[2] transferred between buses $i$ and $j$. Thus, the total inequality constraints $G$ are given by the union of the constraints defined above:

$$G = G_P \cup G_Q \cup G_V \cup G_S \tag{4}$$

We implement the model using the `MATPOWER` Zimmerman et al. (2020) package, which employs the MATPOWER Interior Point Solver (MIPS) Zimmerman & Wang (2016) solver, currently one of the most widely-used solvers for AC-OPF.

## 2.2 Physics Informed Neural Networks with Hard Constraints

Physics-Informed Neural Networks (PINNs) were first introduced in Raissi et al. (2019) as a data-driven approach to solving problems governed by partial differential equations (PDEs). PINNs leverage automatic differentiation to compute high-order derivatives of the neural network, which is treated as the solution of a PDE, with respect to its inputs. Specifically, PINNs aim to learn the solution $u : I \times \Omega \subseteq \mathbb{R}^m \to \mathbb{R}^n$ of a differential problem $\mathcal{F}[u(\mathbf{x}, t)] = 0$ for $\mathbf{x} \in \Omega$ subject to suitable boundary and initial conditions, using a neural network $u_{\text{NN}}$. The network is trained by minimizing the residuals $\mathcal{L}_{\mathcal{F}}$ associated with each equation in the differential problem. PINNs have been widely applied across various domains, including heat transfer problems Cai et al. (2021b), fluid dynamics Cai et al. (2021a), and power systems Misyris et al. (2020). Several variations of PINNs have been proposed to address issues such as structural instability Mai et al. (2023), improve accuracy Eshkofti & Hosseini (2023), or better approximate discontinuous solutions to hyperbolic equations De Ryck et al. (2024).

Physics-informed neural networks with hard constraints (hPINNs) were introduced in Lu et al. (2021) to optimize an objective function $\mathcal{J}$ while incorporating equality constraints $h_j, \forall h_j \in H$ and inequality constraints $g_l, \forall g_l \in G$. This approach uses both the penalty method and the Augmented Lagrangian (AL) method. At each iteration $k$, it minimizes the loss function to identify the solution parameters $\mathbf{w}_u$:

$$\mathcal{L}^k(\mathbf{w}_u^k) = \mathcal{J}(\mathbf{w}_u^k) + \mu_H^k h(\mathbf{w}_u^k)^2 + \mu_G^k g(\mathbf{w}_u^k)^2 + \frac{1}{L} \sum_{l=1}^{L} \lambda_{h_l}^k |h(\mathbf{w}_u^k)| + \frac{1}{J} \sum_{j=1}^{J} \lambda_{g_j}^k max(0, g(\mathbf{w}_u^k)) \tag{5}$$

where the adaptive coefficients $\mu_G^k, \mu_H^k$ are increased by a factor $\beta$ at each step. The Lagrange multipliers $\lambda_{h_l}^k, \lambda_{g_j}^k$ are updated based on the direction of the gradients $\nabla h_j$ and $\nabla g_l$ Lu et al. (2021).

## 2.3 Graph Neural Networks

GNN Scarselli et al. (2008) are a class of machine learning models designed to process structured data. Since their introduction, they have been successfully applied in various domains, such as social networks Li et al. (2023), traffic networks Jiang & Luo (2022), and molecular dynamics Park et al. (2024). GNN can handle both *graph-level* and *node-level* tasks by updating the features of each node. In addition, they can apply a final learnable layer to aggregate the node features and make a prediction for the entire graph. Most GNN proposed in the literature share a common update mechanism known as *message passing*. As described in Gilmer et al. (2017), the message-passing update step in a GNN can be written as:

$$\mathbf{x}_i^{(k+1)} = \mathsf{COMBINE}^{(k+1)} \left( \mathbf{x}_v^{(k)}, \mathsf{AGGR}^{(k+1)} \left( \left\{ \mathbf{x}_u^{(k)} \mid u \in \mathsf{ne}(i) \right\} \right) \right), \ 0 \leq k \leq K - 1 \tag{6}$$

where $\mathbf{x}_i^{(k)}$ represents the features of node $i$ at iteration $k$, and $\{\mathsf{COMBINE}^{(k)}\}_{k=1,\dots,K}$ and

---

[2]Apparent power $S$ in an AC circuit is the product of the RMS voltage and the RMS current, expressed as $S = V \times I^*$, or in terms of magnitude as $|S| = \sqrt{P^2 + Q^2}$, where $P$ is the real power and $Q$ is the reactive power.

$\{\mathsf{AGGR}^{(k)}\}_{k=1,...,K}$ are families of functions defined for each iteration up to depth $K$. Intuitively, the message-passing mechanism first collects information from the neighborhood of each node, denoted as $\mathsf{ne}(i)$, using the function $\mathsf{AGGR}^{(k)}$. The neighborhood $\mathsf{ne}(i)$ refers to the set of nodes connected to node $i$. The aggregated information is then combined with the existing information of node $i$ via $\mathsf{COMBINE}^{(k)}$. Finally, depending on the task, a $\mathsf{READOUT}$ function is applied.

GNN have been shown to be universal approximators for both graph-level and node-level tasks Azizian & Lelarge (2020); D'Inverno et al. (2024); Loukas (2019). We choose GNNs for our problem because they can effectively handle different topologies, in principle allowing a single model to be trained and applied across various power grids Varbella et al. (2023). GNN are also capable of adapting to variations in the topology itself, making them well-suited for the power system optimization Piloto et al. (2024).

## 2.4 MODEL DEFINITION

The concepts and methods in Sections 2.2 and 2.3 are combined to solve the AC-OPF problem. We define the specific input and output structure in Section 3.1. Most importantly, each input sample is defined by $(P_d, Q_d)$, corresponding to a different loading condition of the power grid. The other grid parameters remaining constant are the generation limits, voltage limits, branch admittances, and branch thermal limits. The output at the $k$-th training iteration is denoted by $(P_g^k, Q_g^k, V^k, \theta^k)$.

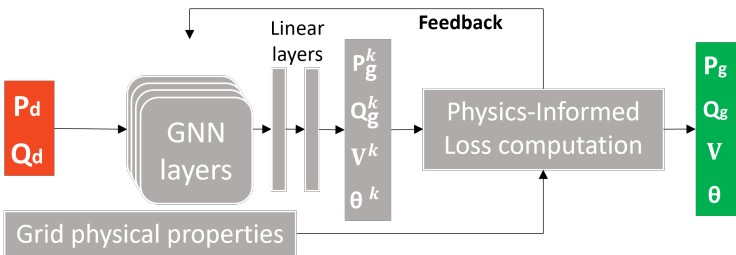

Figure 1: Schematic of the PINCO method architecture. This diagram illustrates the input data (in red) and the output generated by the neural network (in green). The constant parameters specific to the power grid, as defined by the optimization problem in Section 2.1, are fed directly into the physics-informed loss function.

The original formulation of the AC-OPF can be included in a single expression, using Equation 5. This objective function is used as the loss function for the algorithm. By combining a GNN architecture with a physics-informed loss function, we develop a model for AC-OPF called PINCO that uses a Physics-Informed GNN for hard constraints.

## 3 EXPERIMENTS ON THE IEEE BENCHMARKS

The IEEE power system test cases are a widely recognized set of benchmarks frequently employed in power systems research. These test cases represent power grids with their respective characteristics and provide example demand scenarios. In this work, we apply our algorithm to the IEEE 9-bus, IEEE 24-bus, IEEE 30-bus, and IEEE 118-bus cases.

The IEEE 9-bus system Zimmerman et al. (2020), being one of the smallest test cases, includes 3 generator units, 3 loads, and 9 buses, making it a simple yet useful case for foundational testing. The IEEE 24-bus case Texas A&M University Engineering (2022b) has 32 generator units, with some buses containing up to 6 generators. After applying the node-splitting model (as explained in Section 3.1), this effectively becomes a 56-bus system with transformers and parallel lines, making it a small but intricate system ideal for demonstrating the model's robustness. The IEEE 30-bus system comprises 6 generator units, 41 transmission lines, and 4 transformers Christie (1993). The largest test system we use is the IEEE 118-bus system, with 19 generators and 186 transmission lines and transformers Texas A&M University Engineering (2022a).

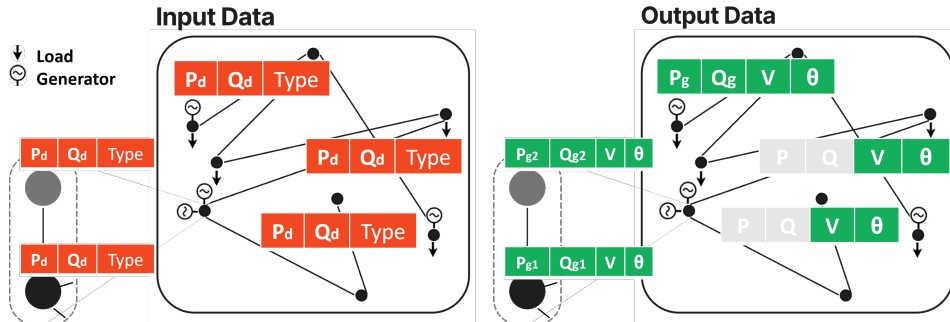

Figure 2: Power grid model as graph. The input features assigned to each node are red, and the predicted quantities at the node level are green. The zoom visualizes how multiple generators per node are modeled; the grey node is the artificial node created to account for additional generators at the node.

### 3.1 DATASET STRUCTURE

We model the electrical grid as a graph, with a set of nodes $N$ corresponding to electrical buses, and edges $E$ to model the branches. The graph is defined as $G = (N, E, \mathbf{N}, \mathbf{A})$. We call $\mathbf{N} \in \mathbb{R}^{|N| \times t}$ the node feature matrix, with $|N|$ equal to the number of nodes and $t$ to the number of features per node, and $\mathbf{A} \in \mathbb{R}^{|N| \times |N|}$ is the adjacency matrix. The elements of $\mathbf{A}$, $a_{ij}$, are equal to 1 if there is an edge from node $i$ to node $j$, and zero otherwise. We predict $\mathbf{Y} \in \mathbb{R}^{|N| \times f}$, the node output matrix; thus, a vector of $f$ element is predicted for each node in $N$.

Figure 2 illustrates the structure of a power grid with its features. The input node features are depicted in red, while the output node-level predictions are in green. The input features include active power demand ($P_d$), reactive power demand ($Q_d$), and node type. The output features vary depending on the node type. For example, if a certain variable is known (e.g., buses without generators don't require predictions for generated power), we apply a masking process during training. These masked values are represented by grey cells. The predicted output quantities include active power generation ($P_g$), reactive power generation ($Q_g$), voltage magnitude ($V$), and voltage angle ($\theta$).

We propose a method to manage buses that have multiple generators. Since each generator is associated with a unique cost, distinguishing between them is essential. To address this, we add an additional node for each extra generator. These nodes are 'artificial,' as they do not correspond to an actual bus on the grid. For example, a bus with two generators, as shown in Figure 2, would be split into two distinct nodes, each linked to its own generator and connected via artificial lines. The voltage magnitude and angle of these artificial nodes are set to match those of the original node. In total, there are four possible node types, encoded as categorical variables: (1) load bus without a generator, (2) bus without load or generators, (3) original generator bus, and (4) artificial generator bus.

### 3.2 EVALUATION METRICS

The proposed method is unsupervised, aiming to solve a non-convex optimization problem. Given that even advanced solvers, such as MIPS, offer no guarantee of optimality for such problems, it becomes crucial to consider alternative metrics for assessing the model's performance. Two fundamental metrics are typically used: (1) the total cost of the solution, which reflects the objective function of the optimization problem, and (2) limit violations, indicating any inequality or equality constraint violations (see Section 2.1). In addition, we introduce a performance metric, which measures the total amount of power deficit in the system. Thus, we quantify the deviation from satisfaction of the equality constraints as follows:

$$eq_{loss} = \sum_{S \in \{P,Q\}} \sum_{i \in N} \sum_{ij \in E} |S_i^{gen} - S_i^{load} - s_{ij}| \tag{7}$$

This metric captures the power deficit at each node, adhering to the principle of energy conservation, and is commonly referred to as the system's "equality loss." Notably, this value is always non-zero,

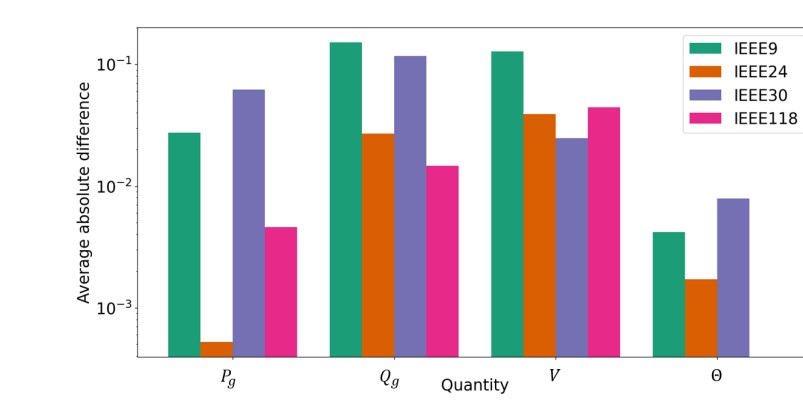

Figure 3: Average differences between solutions from PINCO and the MIPS solver on a logarithmic scale. Phase angle ($\theta$) and voltage magnitude ($V$) differences are averaged across all nodes and normalized by their maximum values. Active ($P_g$) and reactive power ($Q_g$) differences are averaged at generator buses and normalized by total demand.

even for state-of-the-art solvers like the MIPS solver, as solving the AC-OPF problem exactly is typically computationally challenging and infeasible for large systems.

### 3.3 EXPERIMENTAL SETTINGS

All model experiments are conducted on the CPU nodes of the ETH Euler clusters (CSCS). Details regarding the hyperparameters used in these experiments and their selection process are in A.1.

## 4 RESULTS

In this section, we evaluate the PINCO method of the various IEEE benchmark power systems. First, we assess its performance as a solver for a single demand scenario in Section 4.1. Then, in Section 4.2, we test the method's hybrid capability to generalize across multiple demands, demonstrating its role as a universal function approximator while simultaneously solving the optimization problem. Our approach consistently achieves solutions with zero inequality constraint violations, rendering the need for an inequality violation-based metric unnecessary.

### 4.1 PINCO AS A SOLVER FOR A SINGLE LOADING CONDITION

We initially performed experiments on a single instance, using the default demand specified in IEEE benchmark cases. These tests aimed to evaluate the method's potential as a viable alternative to traditional nonlinear optimization solvers. Specifically, they served as a proof of concept to assess whether the algorithm could produce solutions competitive with those generated by state-of-the-art solvers like MIPS, which was used as a reference benchmark.

Figure 3 presents the average differences between the solutions obtained by PINCO and the MIPS solver, displayed on a logarithmic scale. The absolute differences in phase angle ($\theta$) and voltage magnitude ($V$) are averaged across all nodes and divided over the maximum $V$ and $\theta$ values. The absolute differences in active and reactive power ($P_g$ and $Q_g$) are averaged only at buses with generators and normalized over the total active and reactive demand. It is important to note that in the case of IEEE118, there is no reference node, i.e. slack bus, which allows for arbitrary shifts in phase angle between the model and solver. As a result, phase angle comparisons for this case were omitted. While the MIPS solver and our solution may differ, this does not imply that either is incorrect.

Table 4.1 provides a more comprehensive comparison by presenting the equality losses for both methods, with our model's results listed under 'PINCO equality loss' and the MIPS results under

Table 1: Test results for single input demand profile using the custom evaluation metrics as defined in section 3.2.

| Power Grid | PINCO Equality Loss [MW] | MIPS Equality Loss [MW] | Rel Cost Difference [%] |
|---|---|---|---|
| IEEE9 | 0.003 | 0.002 | 1.10 |
| IEEE24 | 0.040 | 6.500 | 0.63 |
| IEEE30 | 0.018 | 0.015 | 4.90 |
| IEEE118 | 0.067 | 20.000 | 1.20 |

'MIPS equality loss.' In all cases, the cost differences are positive, indicating that our method consistently yields solutions that are slightly more expensive than MIPS.

For the simpler IEEE9 and IEEE30 cases, the model is able to produce solutions with similar equality losses and costs, meaning it finds physically feasible solutions, though they may be suboptimal in terms of cost. In the more complex IEEE24 and IEEE118 cases, it is noteworthy that the equality loss for MIPS is unexpectedly high. This highlights a key challenge in non-convex optimization: the solution is heavily influenced by the objective function's formulation. The MIPS solver tends to focus on minimizing costs, even if that results in higher equality losses, especially when navigating a complex solution space. Conversely, our method prioritizes respecting equality constraints and finds solutions that are physically more accurate, while slightly more costly, i.e., 0.6% for IEEE24 and 1.2% for IEEE118.

### 4.2 PINCO AS UNIVERSAL FUNCTION APPROXIMATOR ON MULTIPLE LOADING CONDITIONS

After evaluating PINCO's performance as a solver, we proceeded to test its ability to function as a universal function approximator, capable of generalizing to unseen demand conditions while solving the problem directly without the need for labeled data. For a given reference loading condition, the active and reactive power demands are sampled from a uniform distribution around 90% and 110% of their reference values. For each experiment, we generate 500 input demand samples. Therefore, the dataset for multiple loading conditions consists of $W$ attributed graphs, denoted as $\mathcal{G} = G_1, G_2, \ldots, G_W$, with each graph representing a different power grid loading condition. For all cases, the training, validation, and test datasets were created from a common set of generated inputs, which were then split respectively into 80%, 10%, and 10%. To evaluate PINCO's generalization capability in addressing the AC-OPF under unseen loading conditions, we assess its performance on the test set and compare the results with those obtained from MIPS, utilizing the metrics detailed in Section 3.2.

Table 2: Results for multiple loading conditions on the test set.

| Power Grid | PINCO Equality Loss [MW] | MIPS Equality Loss [MW] | Cost Difference [%] |
|---|---|---|---|
| IEEE9 | 0.030 | 0.001 | 0.010 |
| IEEE24 | 4.300 | 6.500 | 0.880 |
| IEEE30 | 0.690 | 0.020 | 0.800 |
| IEEE118 | 16.000 | 20.000 | 1.100 |

Table 4.2 shows that our method produces solutions that are comparable to those of the MIPS solver. While the equality loss is higher by an order of magnitude for simpler cases like IEEE9 and IEEE30, PINCO achieves better equality losses for the more complex IEEE24 and IEEE118 cases, as seen also in the previous Section 4.1. Across all test cases, the model's solutions exhibit only around a 0.8% increase in cost compared to the solver's solution. Moreover, even in instances where PINCO underperforms, it offers a valuable trade-off due to its significant improvements in inference times (see Figure 4), which were obtained using the same setup. Our method in the inference phase is two orders of magnitude faster than MIPS.

## 5 LIMITATIONS

Despite its strengths, the PINCO method exhibits some limitations. While inference is highly efficient, training the model remains computationally expensive, which can be a barrier to broader application. Depending on the grid size, the training can last from 10 to 24 hours. Additionally, when the model is trained on multiple demand scenarios, its ability to respect equality constraints

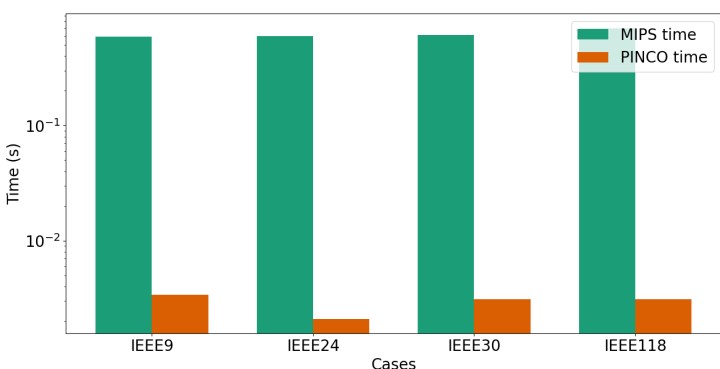

Figure 4: MIPS and PINCO inference times in logarithmic scale, tested using the same conditions on the same device (CSCS).

requires further refinement to improve overall performance. Another challenge is hyperparameter tuning. While we achieved results with minimal tuning (refer to A.1), the impact of hyperparameters warrants further investigation. Addressing these limitations will be crucial in fully leveraging the potential of PINCO for power grid optimization tasks.

## 6 CONCLUSIONS

In this study, we develop a physics-informed graph neural network-based method, namely PINCO. We evaluate the performance of the PINCO method to solve an optimization problem with hard constraints, i.e., the AC-OPF. We assessed its capability as a solver for a single-demand scenario and observed that it generalizes well to multiple demands. Thus, showcasing its role as a universal function approximator that solves optimization problems.

The method is tested on multiple IEEE benchmark systems, representing different grid topologies and sizes, with a focus on grids containing multiple generators. Across all tests, the PINCO demonstrates zero inequality constraint violations, highlighting its effectiveness in handling hard constraints. When applied as a solver, the PINCO provides solutions comparable to traditional solvers for the IEEE 9-bus and IEEE 30-bus systems. Moreover, it outperforms traditional solvers in reducing equality constraint violations for the more complex IEEE 24-bus and IEEE 118-bus systems, though this comes at a marginal cost increase of only 0.6% and 1.2%, respectively.

In scenarios where the PINCO is tested on multiple demands, its generalization performance for unseen loading conditions shows a slight deterioration in the equality constraint violations, while the associated costs remain comparable to traditional methods. However, the method's computational advantage in the inference phase, being two orders of magnitude faster than traditional solvers, makes it a highly competitive alternative. Future work will focus on scaling the method to larger grids with more realistic loading conditions and addressing the N-1 AC-OPF problem in an unsupervised manner.

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

# A APPENDIX

## A.1 HYPERPARAMETERS

We report here all the hyperparameters used for the experiments. In Table 3, we present the hyperparameters for the experiments where a single loading condition is concerned. In this case, the architecture included 5 TransformerConv Shi et al. (2020) layers followed by two linear layers with tahnshrink [3] activation function. The hidden dimension is set to 24, with 4 attention heads in the transformer layer.

Table 3: Experimental set up for experiments where the method is used as a solver, i.e. a single loading condition is considered. Hyperparameters and training conditions are reported

| Power grid | Case9 | Case24 | Case30 | Case118 |
|---|---|---|---|---|
| $\mu_g$ | 0.001 | 0.001 | 0.001 | 0.001 |
| $\mu_h$ | 0.1 | 0.1 | 0.1 | 0.1 |
| $\beta_g$ | 1.00002 | 1.00002 | 1.00002 | 1.00002 |
| $\beta_h$ | 1.00003 | 1.00003 | 1.00003 | 1.00003 |
| Epochs | 200000 | 200000 | 200000 | 200000 |
| Learning rate | 0.0005 | 0.0005 | 0.0005 | 0.0005 |
| $\gamma$ | 0.9995 | 0.9995 | 0.9995 | 0.9995 |

In Table A.1, we report the ones for multiple loading conditions. In this case, the architecture is the same as the one used for a single loading condition, but 8 TransformerConv layers are employed. A batch size of 20 is used for these experiments.

Table 4: Experimental set up for experiments where the method is used as a universal function approximator, i.e., 500 loading conditions are considered. Hyperparameters and training conditions are reported

| Power grid | Case9 | Case24 | Case30 | Case118 |
|---|---|---|---|---|
| $\mu_g$ | 0.001 | 0.001 | 0.001 | 0.001 |
| $\mu_h$ | 0.2 | 0.1 | 0.1 | 0.1 |
| $\beta_g$ | 1.00002 | 1.00002 | 1.00002 | 1.00002 |
| $\beta_h$ | 1.00003 | 1.00005 | 1.00005 | 1.00005 |
| Epochs | 160000 | 160000 | 160000 | 160000 |
| Learning rate | 0.0005 | 0.0005 | 0.0006 | 0.0005 |
| $\gamma$ | 0.9995 | 0.9995 | 0.9995 | 0.9996 |

The models are trained using an initial learning rate, as shown in Tables 3 and A.1, with an exponential learning rate schedule that reduces the learning rate by a factor, $\gamma$, every ten epochs. While the initial learning rate does not significantly impact the training results, selecting a small value for $\gamma$ can slow down the learning process, causing the optimizer to prematurely converge to a local minimum. The hyperparameters related to the h-PINN method Lu et al. (2021), such as $\mu_g$, $\mu_h$, $\beta_g$, and $\beta_h$, were chosen through a grid search, centered around the values suggested in the original paper.

---

[3]Tanhshrink$(x) = x - \tanh(x)$