# OpenReview forum: "Physics-informed GNN for non-linear constrained optimization: PINCO, a solver for the AC-optimal power flow"
_ICLR.cc/2025/Conference — Submitted to ICLR 2025_

### Official Review · Reviewer_UGxj · 2024-10-15

**Soundness:** 2
**Presentation:** 1
**Contribution:** 1
**Rating:** 1
**Confidence:** 5

**Summary:**

This paper presents a graph neural network architecture for solving AC Optimal Power Flow models.
The paper describes a physics-informed loss for training the model in a self-supervised manner, and conducts numerical experiments on systems with up to 118 buses.

**Strengths:**

The proposed methodologies has two positive aspects
* it uses a graph-based architecture, which supports changes in grid topology (however this capability was not demonstrated)
* it uses a self-supervised training scheme that does not require the (expensive) use of an optimization solver to generate data

However, neither of these are new, and the paper suffers from multiple limitations (see below).

**Weaknesses:**

Limitations of the methodologies / experiments:

* The paper incorrectly states that previous works do not handle multiple generators per bus.
  * _Confidence-Aware Graph Neural Networks for Learning Reliability Assessment Commitments_ (https://doi.org/10.1109/TPWRS.2023.3298735) consider graph neural network architectures for unit-commitment problems, and present an encoder-based mechanism for handling multiple generators per bus
  * _CANOS: A Fast and Scalable Neural AC-OPF Solver Robust To N-1 Perturbations_ (https://arxiv.org/abs/2403.17660) does support such features:
    _We also use artificial edges to connect the subnodes to their respective bus (these do not model any physical equipment)._
  * It should be noted that handling multiple generators at a single bus is not a hard task, as they can easily be aggregated into a single generator; an aggregated solution can then be disaggregated in closed-form.

* I believe the augmented Lagrangian in Eq. (5) is incorrect, specifically the term $\mu_{k} g_{j}(w_{u}^{k})^2$.
  Assuming inequality constraints are of the form $g(w) \leq 0$, this quadratic term effectively drives all inequality constraints towards being binding (i.e. $g(w) = 0$), which is not theoretically sound.

* The paper argues that the use of a graph neural network supports changes in the grid topology, but does not present experiments that corroborate this claim

* The paper claims that the proposed model can achieve "zero violation of inequality constraints." This is not a substantial achievement, given that said inequality constraints are either simple variable bounds (which can be enforced via a bounded activation function) or simple l2 norm constraint (which can also be enforced by a simple scheme). The complexity of AC-OPF is in satisfying _both_ equality and inequality constraints. It is straightforward to enforced either set of constraints separately (i.e. only equality or only inequality constraints).

* Numerical experiments consider AC-OPF instances on systems with up to 118 buses. This is 100x smaller than real-life instances, which comprise (at least) in the order of 10,000 buses.

  It should be noted that several works have trained ML models to predict AC-OPF with systems of that scale:
  * _Spatial Network Decomposition for Fast and Scalable AC-OPF Learning_ (https://doi.org/10.1109/TPWRS.2021.3124726)
  * _Compact optimization learning for AC Optimal Power Flow_ (https://arxiv.org/abs/2301.08840): up to 30,000 buses
  * _CANOS: A Fast and Scalable Neural AC-OPF Solver Robust To N-1 Perturbations_ (https://arxiv.org/abs/2403.17660): up to 10,000 buses, and also uses graph neural network architectures

* Training times appear to be very large (10 to 24 hours as reported in Section 5). Given that the paper only considers very small artificial systems, it is not clear that the proposed scheme would scale to real-life systems. A reasonable target would be at most 6-8hrs of training time on a systems with about 10,000 buses.

* Numerical experiments do not compare against any existing ML methodology for AC-OPF problems.

Issues about the paper:
* Not all notations are properly defined. For instance, the formalism of equality and inequality constraints is not defined before the Augmented Lagrangian presented in Eq. (5). Similarly, this equation uses a parameter $w^{k}_{u}$ that is not defined.
* the paper does not present the data distribution used to train the model. It should be noted that generating data by perturbing individual loads independently is not realistic, as it leads to a very narrow variability in total demand.

  Note that there are open-source datasets and data generators for AC-OPF problems, e.g.:
  * OPFData (https://arxiv.org/html/2406.07234v1) is an open-source dataset of AC-OPF instances and their solutions
  * OPFGenerator (https://github.com/AI4OPT/OPFGenerator) is an open-source instance generator for various OPF formulations

* In addition to the comments, above, the paper sometimes makes misleading claims. For instance, at lines 88-89, it is stated that "INCO allows for solving the AC-OPF **without violations**." This claim is not substantiated by the results reported in Section 4 (Tables 1 and 2).

**Questions:**

In the absence of a fundamentally new architecture or training scheme, to be accepted, this paper should:
* conduct numerical experiments on systems with at least 10,000 buses...
* ... with a more realistic data distribution than independently perturbing individual loads;
* demonstrate the proposed architecture's ability to handle varying topologies;
* achieve training times of at most 8hrs (this is the kind of timeline that would be needed for practical applications);
* achieve low constraint violation (in the order of $10^{-4}$ relative violation) and optimality gap (no more than $0.1$% worse than Ipopt);

It should be noted that each of the above bullet has been addressed in the literature, at least individually.
For instance, there exist works that scale to 10,000-bus systems, works that consider more realistic data distributions, works that achieve low constraint violations, etc...

---

> ### Author Response · Authors · 2024-11-22
> **Reply to reviewer UGxj**
>
> We appreciate the detailed feedback provided; while some concerns highlight valid areas for clarification and improvement, we believe other critiques may stem from misunderstandings or misalignment of expectations regarding the goals of our work.
> Our focus is on solving the AC-OPF problem in a fully unsupervised manner using a combined GNN+PINN framework, differentiating this work from others relying on supervised learning techniques. We believe this addresses many of the comments regarding comparisons with prior works. While we have decided not to proceed further with the submission, we would still like to take this opportunity to clarify the points raised.
>
> 1. While the referenced works address multiple generators per bus, they rely on supervised learning approaches or tackle entirely different problems (e.g., unit commitment). Our approach, by contrast, solves the AC-OPF problem in a fully unsupervised manner. Additionally, we stress that aggregating and disaggregating generators, as suggested by the reviewer, introduce inaccuracies. In practical settings, operators require detailed generator-level outputs—not aggregated approximations. Particularly, how do we deal with generators but with different costs?
>
> 2. We disagree with this claim and invite the reviewer to refer to the references [1] [2]. Both references provide theoretical backing for the Augmented Lagrangian approach employed in our work. We want to ask if you could elaborate on why achieving inequality constraints equal to zero is not theoretically sound.
> [1] Lu et Al., "Physics-Informed Neural Networks with Hard Constraints for Inverse Design"
> [2] Toussaint M., "Introduction to Optimization"
>
> 3. We acknowledge the lack of experiments on this aspect. In the next version of the manuscript, we will include a case study demonstrating the proposed model's ability to handle topology variations.
>
> 4. While we agree that enforcing inequality constraints can sometimes be straightforward, we emphasize that prior fully unsupervised approaches have not achieved this in the AC-OPF context. Our work demonstrates zero inequality violations in a fully unsupervised setting, a non-trivial achievement. Additionally, the results on equality constraint violations are comparable to traditional solvers, as reported in Tables 1 and 2.
>
> 5. We disagree for several reasons:
>
> Realism of Test Systems:
>
> - Many real-world transmission systems, such as the Swiss transmission grid, comprise fewer than 200 nodes (e.g., 186 nodes in the Swiss system, 3657 substations for the European transmission systems https://arxiv.org/abs/1806.01613).
>
> - A more extensive test case like the one presented in https://arxiv.org/abs/2301.08840 is synthetic and does not reflect the structure of actual grids.
>
> Comparison with the references reported:
>
> - Unlike the supervised methods reported, our approach does not require labeled data from traditional solvers.
>
> - While we acknowledge the importance of scalability, this work focused on validating the framework on standard benchmarks.
>
> 6. Training times are a standard limitation of any deep learning framework. Training is a one-time process; the benefit always lies in the fast inference times once the model is trained, which is crucial for real-time applications.
>
> 7. Direct comparisons with supervised ML methodologies are not meaningful because our approach is fully unsupervised. Only one work was found in [3], which presented inequality constraint violations. Supervised approaches approximate solutions from traditional solvers, while our method directly solves the problem, finding alternative feasible solutions.
> [3] https://arxiv.org/abs/2210.09277
>
> 8. We respectfully disagree, as the parameter is defined in the text immediately preceding Eq. (5).
>
> 9. We acknowledge that using a more realistic data distribution could improve the model's generalizability. However, our current approach (uniform distribution) is explicitly stated in Section 4.2. This method is commonly used in literature [3],[4].
> [4] "PGLearn - An Open-Source Learning Toolkit for Optimal Power Flow.
>
> 10. We agree that the phrasing could be clearer. Our claim specifically refers to inequality violations, and we will revise the future text to reflect this distinction explicitly.
>
> 11. We appreciate the final suggestions and offer the following responses:
> - 10,000-Bus Systems: Testing on such large systems is a valuable future direction, but we note that real-world grids often contain fewer nodes (e.g., Swiss grid, European Transmission System).
>
> - Realistic Data Distributions: This will be explored further in future work.
>
> - Topology Changes: Experiments will be included in the next version accounting for topology variations.
>
> - Training Times: While deep learning frameworks inherently require significant training times, we will explore optimizations.
>
> - Comparisons: Direct comparisons with ML methods are challenging due to different problem formulations (unsupervised vs. supervised).

---

> > ### Comment · Reviewer_UGxj · 2024-11-26
> >
> > 1. My point is that i) other works have proposed architectures that consider multiple generators per bus, and ii) in the OPF setting, it is straightforward to aggregate/disaggregate multiple generators per bus.
> >
> >    For instance: consider two generators with output $p_{1}, p_{2}$, minimum output $0$ and maximum output $p_{1}^{max}, p_{2}^{max}$ and cost $c_{1}, c_{2}$. An aggregate model will predict the aggregate power $\bar{p} = p_{1} + p_{2}$, with corresponding maximum limit $\bar{p}^{max} = p_{1}^{max} + p_{2}^{max}$. Given a predicted $\bar{p}$, and assuming without loss of generality that $c_{1} < c_{2}$ it is then immediate to recover $p_{1} = min(\bar{p}, p_{1}^{max})$ and $p_{2} = \bar{p} - p_{1}$. More generally, the disaggregation step can be performed in parallel for every bus, and can support more general (e.g. quadratic of piece-wise linear) cost functions and arbitrary number of generators.
> >
> > 2. Given inequality constraints of the form $g(x) \leq 0$, and ignoring equality constraints for simplicity here, the augmented Lagrangian as stated in the paper has the form $L_{\mu}(x, \lambda) = J(x) + \lambda \max(0, g(x)) + \mu g(x)^{2}$ where $\mu > 0$ and $\lambda$ is the Lagrange multiplier. Now consider the problem $\min_{x \leq 0}$, which yields $L_{\mu}(x, \lambda) = x + \lambda \max(0, x) + \mu x^{2}$. The original problem is unbounded, yet the (augmented) Lagrangian problem always has a finite solution; this is contradictory because $\min_{x} L_{\mu}(x, \lambda)$ should always be a lower bound on the optimal value of the original problem.
> >     I think the original form of Eq. (5) was a typo, but I would encourage the authors to consider the augmented Lagrangian formulation used in, e.g., Lancelot.
> >     (Note: I chose an unbounded linear problem for simplicity, other examples can be constructed that are bounded and have strictly convex objective).
> >
> > 3. N/A
> > 4. The point is that, for AC-OPF, what is hard is to satisfy _both_ equality and inequality constraints. The inequality constraints in AC-OPF involve only i) variable bounds on active/reactive generation and voltage magnitude, which can be enforced using a scaled/shifted sigmoid activation and ii) convex quadratic constraints $p_{ij}^{2} + q_{ij}^{2} \leq S_{ij}$, which are also easy to enforce using, e.g., a re-parametrization or a closed-form re-scaling (see, e.g., papers like RAYEN).
> >     As far as I can tell, the proposed methodology does not guarantee both equality and inequality constraint satisfaction, which I think is a substantial limitation of the method for AC-OPF problems. If the authors want to showcase their methodology on general, non-trivial inequality constraints, then I would recommend presenting experiments on multiple classes of problems (where inequality constraint satisfaction should be non-trivial).
> >
> > 5. The PyPSA European grid is synthetically reconstructed (the original paper reports, for instance, that all line impedance information is artificial). The PEGASE project released snapshots of the european grid, and these snapshots have up to 13,000 buses (see https://arxiv.org/abs/1603.01533). In the same paper, several snapshots of the RTE system were released, all of which have over 6,000 buses.
> >
> > 6. I agree that training time is usually considered a "sink cost" in the ML literature. However, if one can spend arbitrary time training a model, then it should be OK to also spend time generating data for supervised learning. Therefore, besides the final performance of the model, the comparison between supervised vs self-supervised approaches should include [data-generation time + training time] as a metric. As an example: for similar accuracy levels, spending {10hrs data generation + 2 hours supervised training} is better than {0hrs data generation + 16hrs self-supervised training}.
> >
> > 7. Supervised and self-supervised methods both aim to address the same problem, and should therefore be compared. As mentioned above, the comparison should include not only overall accuracy but also data-generation, training and inference time.

---

### Official Review · Reviewer_gXzX · 2024-10-31

**Soundness:** 3
**Presentation:** 3
**Contribution:** 2
**Rating:** 3
**Confidence:** 4

**Summary:**

This paper presents PINCO, a neural architecture for solving the AC-OPF problem. PINCO is a combination of GNN + PINNs with hard constraints. Numerical tests show the usefulness of the proposed model.

**Strengths:**

+ A new model combining the GNNs and PINNs for solving AC-OPF
+ Promising numerical performance relative to MIPS solver

**Weaknesses:**

- Both GNNs and PINNs have been used for solving AC-OPF, and the paper directly combines the two without much novel design in the architecture.
- Insufficient comparison against other numerical solvers for AC-OPF of distribution systems including e.g., using GNNs, PINNs, SDP relaxation and SOCP relaxation, and linear-OPF initialized Newton-Raphson method in terms of performance.

**Questions:**

Q1. Any specific design in the GNN+PINN architecture?
Q2. Solvers using GNNs or PINNs shall be included as baselines to show the usefulness of having both in a single architecture.
Q3. It is not clear why the proposed PINCO would improve upon the PINNs with hard constraints approach? This shall be clarified as well as evidenced using numerical comparison against a number of benchmark test systems.
Q4. Physics-informed GNN based power system state estimation (PSSE) was among the first use of NN architecure for power systems and they shall be discussed as a motivation of using deep learning for power systems.

---

> ### Author Response · Authors · 2024-11-22
> **Reply to reviewer gXzX**
>
> We appreciate the time and effort the reviewer has dedicated to evaluating our work. However, we find it challenging to reconcile the overall tone of the feedback, which acknowledges the paper’s soundness, presentation, and contributions, with a relatively low rating of 3. We kindly ask the reviewer to clarify this discrepancy. While we have decided not to proceed further with the submission, we would still like to take this opportunity to clarify and address the points raised.
>
> 1. The architecture employed in our work was carefully customized for the AC-OPF problem by leveraging graph transformer layers, as detailed in the Appendix. While we acknowledge that the individual components (GNNs and PINNs) are established techniques, their combination in this context is novel. Importantly, we emphasize that our contribution lies not in introducing unnecessary architectural complexity but in demonstrating that the proposed framework, built on graph-transformer-based GNNs and a PINN-based unsupervised learning approach, can achieve zero inequality violations and generalize to unseen scenarios. Moreover, to the best of our knowledge, this work is the first to successfully apply the H-PINN framework to the AC-OPF problem, a key innovation that we believe adds significant value. This type of PINN is designed for problems with hard constraints, such as the OPF, and to our knowledge, we are the first to apply this method successfully in this field.
>
> 2. We thank the reviewer for this comment. While we agree that a comparison with another IP solver (e.g., IPOPT in addition to MIPS) could be added, our work is not aimed at addressing every modeling approach to the OPF problem. Instead, we focus on demonstrating that an unsupervised ML approach can solve the full OPF formulation without relying on relaxations or simplifications.
>
> Q1. Yes, our GNN+PINN architecture was specifically tailored for the AC-OPF problem by integrating graph transformer layers to effectively model the power grid's topology. This choice was motivated by prior work indicating the superior performance of transformer-based architectures in power systems. Furthermore, we are the first to apply the H-PINN method to solve the ACOPF problem.
>
> Q2. While we acknowledge the importance of comparisons, methods using GNNs or PINNs in isolation are fundamentally different from our combined approach. Specifically: A GNN-only solver would require supervised learning and depend on labeled datasets generated by traditional solvers, which our method aims to eliminate. A PINN-only solver would not benefit from the structural insights provided by the GNN architecture. The strength of our framework lies in the combination of these, enabling an unsupervised solution that generalizes well. Therefore, while we agree our work could benefit from more comparison with other AC-OPF solvers, it would not be possible to compare it fairly with other ML techniques.
>
> Q3.  We believe the reviewer is requesting clarification on the necessity of using PINNs with hard constraints in the AC-OPF problem. To address this:
>
> - Necessity of Hard Constraints: The AC-OPF problem involves both equality and inequality constraints that are critical for ensuring physical feasibility. Using a PINN with hard constraints ensures that these conditions are met during optimization, making it well-suited for this application.
>
> - Improvement: Our method improves upon standard PINN approaches by integrating a graph-based architecture, which captures the underlying power grid topology more effectively. This, combined with the H-PINN framework, allows us to achieve zero inequality violations, a significant improvement over previous works.
>
> Q4. We appreciate this suggestion and acknowledge the importance of prior work on physics-informed GNN-based state estimation. However, it is important to note that the focus of this paper is on AC-OPF, not state estimation. While these domains are related, they address different challenges within power systems. If the reviewer could provide specific references for the PSSE work and elaborate on why they should be discussed in a work only focused on OPF, we would be happy to review and consider including them as part of our motivation in the revised manuscript.

---

### Official Review · Reviewer_Nm3Q · 2024-11-04

**Soundness:** 2
**Presentation:** 3
**Contribution:** 2
**Rating:** 1
**Confidence:** 5

**Summary:**

This paper considers a physics-informed graph neural network, PINCO, to solve the AC-OPF problem. Using standard test cases, it shows that proposed method is faster than standard nonlinear solvers. Some generalization properties are discussed.

**Strengths:**

- ACOPF is an important problem in grid operations.
- The paper is easy to read, and the results seem correct.

**Weaknesses:**

- In my opinion the main weakness of the paper is that the result isn't that good. In the sense that the main goal of the paper is to find feasible solutions, and I don't think that goal is accomplished.
- In Figure 3, the violations of $P, \theta, Q,$ and $V$ are shown. But the violations can be pretty large, especially for $Q$ and $V$. One of the reasons ACOPF is ran is to handle the V/Q constraints, and having a 10% error is not great.
- Table 2 reports the "equality constraint" violations when the input load scenarios changes. But this violation can be 16 MW, which is again not a small number. I don't think operators would be likely to accept these types of violations. Although the nonlinear solver can be slow, but the problem can be resolved when the load changes.
- The approaches in the paper is based on generic methods, and can be applied to any constrained optimization problem. It would be good to see if there is anything special about the ACOPF when applying the method.

**Questions:**

There has been a lot of work on ACOPF (some are cited in the paper). The authors should compare against some of these, in addition to standard solvers.
What happens when the problem is infeasible? Presumably the neural network would still output something, but would one be able to tell that there is actually no solution?

---

> ### Author Response · Authors · 2024-11-22
> **Reply to Reviewer Nm3Q**
>
> We appreciate the time and effort the reviewer has dedicated to evaluating our work. However, we find it difficult to reconcile the detailed evaluation—characterizing the paper as having fair soundness, good presentation, and a fair contribution—with a final rating of 1. We would kindly ask for clarification regarding this rating, as the provided feedback appears to reflect a more moderate assessment of the paper. While we have decided not to proceed further with the submission, we would still like to take this opportunity to clarify and address the points raised.
>
> 1. All results presented in the paper explicitly demonstrate zero inequality constraint violations, which is a key indicator of feasibility. Additionally, the equality constraint violations achieved by our method are comparable to, and often lower than, those of traditional solvers like MIPS. This is evidenced in the numerical results presented in the manuscript, particularly in Table 2. The reviewer does not provide specific arguments or examples to support this statement. We kindly ask the reviewer to elaborate further so that we can better address their concerns.
>
> 2. We believe the reviewer has misunderstood the intent of Figure 3. This figure does not show constraint violations but rather a comparison of results between our method and the traditional solver (MIPS). The ACOPF problem does not have a unique solution; different solvers may converge to different feasible solutions that represent distinct local minima. These differences often reflect trade-offs between objectives, such as operating cost and strict adherence to power flow equations. Figure 3 highlights such trade-offs. To avoid further confusion, we will revise the figure's description and include additional context in the next version of the manuscript to make its purpose clearer.
>
> 3. The values reported in Table 2 represent the maximum equality constraint violations observed across a test set of 50 grids under varying load scenarios. These values do not imply that the problem is infeasible or that operators would reject the solutions outright.
> For comparison, the equality constraint violations of our method (16 MW) are smaller than those of MIPS (20 MW). Additionally, the violations observed are within acceptable tolerances for large-scale nonlinear optimization problems like ACOPF. To provide additional validation, we plan to include results obtained with other solvers (e.g., IPOPT) in the next version of the manuscript.
>
> 4. As stated in the introduction, the ACOPF problem is of significant importance in power system operations and analysis due to its complexity and practical relevance. Our focus is on applying the proposed framework to this domain. While the method is general and could be extended to other optimization problems, these applications are beyond the scope of the current work.
>
> 5. You raise an important point about comparisons with existing ML methods. However, our method's unsupervised nature makes direct comparisons with most ML approaches (which are supervised) not relevant, as those methods typically map traditional ACOPF solutions (obtained via solvers) and do not solve the problem directly. Instead, it will be more relevant to compare our method to traditional optimization solvers, including MIPS and IPOPT, as both solve the problem directly.
>
> 6. Indeed, the neural network will always output a solution, even when the problem is infeasible. However, infeasibility can be detected by monitoring the violation of inequality constraints or if numerical instability during training is observed. If such violations exceed acceptable thresholds, it indicates that no feasible solution exists. In our experiments, we did not encounter cases of infeasibility, as all presented results respect inequality constraints. We will include infeasibility detection mechanisms in the future version of the manuscript.

---

### Official Review · Reviewer_CnJU · 2024-11-04

**Soundness:** 3
**Presentation:** 4
**Contribution:** 2
**Rating:** 5
**Confidence:** 4

**Summary:**

- This paper introduces PINCO, a physics-informed graph neural network (GNN) designed to approximate solutions for the frequently employed Alternating Current Optimal Power Flow (AC-OPF) problem in power transmission networks. By leveraging existing independent tools like physics-informed NNs and GNNs, PINCO distinguishes itself from prior approaches by better adhering to the inequality constraints of the AC-OPF problem, thereby enhancing solution feasibility in a data-driven framework.

**Strengths:**

- The paper is clear in its exposition and is well-structured that makes the contributions and methodology easy to follow.
- Prior work is clearly acknowledged, and the authors effectively situate their approach within the existing body of research
- The authors make effective use of established methodologies to tackle the AC-OPF problem through an unsupervised learning approach with a focus on ensuring feasible AC-OPF solutions
- Experimental results do highlight feasible AC-OPF solutions quantified via the Equality Loss at the expense of slightly higher cost difference (which is inevitable given the data-driven model).
- The authors also showcase the additional benefits such as faster inference times

**Weaknesses:**

- *Novelty*: The novelty of this approach appears limited, as it primarily relies on well-established tools like GNNs and PINNs. The work could further benefit from clearer differentiation that goes beyond simply combining existing frameworks. For e.g., did the authors explore architectural modifications to the GNN tailored to the characteristics of power transmission networks, or a customized optimization technique designed for AC-OPF? This could significantly enhance the paper’s originality within the ML domain.

- *Scope*: The current approach also presents practical concerns in real-world applications. In practice, power system operators frequently adjust network topology due to maintenance, unplanned outages, and other operational needs, leading to variations in the grid adjacency matrix for a *given* power system test case. Based on the paper’s description, the proposed model would need re-training from scratch for each such new grid adjacency matrix, which presents a notable limitation. Did the authors explore methods to make their model more adaptable to changing topologies for a given test case (even if the cost of the OPF solution is relatively high)? Developing a more flexible framework that can accommodate such variations within a single model would make the approach far more practical than showcasing the performance results across multiple IEEE benchmark systems.

**Questions:**

- Could the authors elaborate on the ML-specific challenges encountered when modeling more than one generator per electrical bus? Is there a fundamental challenge that has prevented past approaches from addressing such cases, or is this simply a modeling choice that the authors address by introducing artificial nodes?

- When solving the IEEE 118-bus problem, any node can be designated as the reference node by including the reference bus angle as an equality constraint in the OPF optimization problem. Could the authors clarify why phase angle comparisons are not included in their evaluation?

- The comparisons in Figure 3 could benefit from additional context. While the differences between the MIPS solver and the proposed solution are shown, it’s not immediately clear how these absolute differences inform the effectiveness of the model, given that the OPF objective is primarily centered on minimizing generator operating costs. Could the authors clarify the purpose of Figure 3 and what specific insights they intended to convey through these comparisons?

- To enhance interpretability, it might be useful to include visualizations that track the evolution of equality loss and relative cost differences throughout training. This could provide more relevant insights into how well the model aligns with the OPF objective over time.

- As noted in the weaknesses, the current formulation focuses on a *single* fixed network topology for each power system case. However, in real-world applications, system operators often encounter dynamic topologies due to changes in operational conditions. Formulating the OPF learning problem to accommodate time-varying grid topologies is more relevant and such a formulation would also give rise to ML modeling challenges that could inspire innovative techniques tailored to real-world needs, potentially making a stronger contribution to both the ML and power systems communities.

---

> ### Author Response · Authors · 2024-11-22
> **Reply to reviewer CnJU**
>
> We thank the reviewer for their thoughtful comments and feedback. While we have decided not to proceed further with the submission, we would still like to take this opportunity to clarify and address the points raised.
>
> Novelty: We appreciate the reviewer’s feedback regarding the novelty of the work. We acknowledge that GNNs and PINNs are well-established frameworks. However, our work is novel in the following ways:
>
> 1.	Application of the H-PINN framework to AC-OPF: To the best of our knowledge, we are the first to successfully apply the H-PINN framework to the AC-OPF problem and achieve zero constraint violations in an unsupervised setting. This type of PINN is designed for problems with hard constraints, such as the OPF, and to our knowledge, we are the first to apply this method successfully in this field.
>
> 2.	Investigation of GNN architecture: While we utilize GNNs, we carefully evaluated different architectures, concluding that the Graph Transformer architecture is the most effective for power systems. This choice stems from prior observations and empirical results, which we highlight in our work.
>
> Scope: We agree with the reviewer that handling topology variations is a critical consideration for real-world applications. While our current work does not explicitly address topology variations, we acknowledge this limitation and plan to address it in future work.
>
> Specifically:
>
> •	Planned evaluation on topology changes: We aim to test our framework on datasets that include N-1 and N-k contingencies. This will help evaluate the ability of the GNN to generalize across different topologies.
>
> •	Adaptability of the GNN model: Incorporating such contingencies will allow us to explore methods to enhance the GNN's adaptability without requiring retraining for each new adjacency matrix.
>
>
> ML-specific challenges in modeling multiple generators per electrical bus: The use of artificial nodes to represent multiple generators per bus is a modeling choice rather than a fundamental limitation. As noted by the reviewer, this adds complexity and requires the framework to learn a more granular solution. We observed this in prior work, where results for grids with multiple generators at the bus level were not observed.
>
>
> Phase angle comparisons in the evaluation: We clarify that the absence of phase angle comparisons stems from the use of the MATPOWER data structure and the MIPS solver, which do not designate a slack node or assign a reference angle for the IEEE 118-bus system.
>
>
> Context in Figure 3: The purpose of Figure 3 is to compare the predicted physical variables (e.g., voltages, powers) of the proposed framework (PINCO) with those of the MIPS solver. While generator operating cost is a central focus in OPF, respecting physical constraints such as grid properties is equally important. The figure primarily aims to illustrate how well the proposed method aligns with these physical constraints. We thank the reviewer for pointing out the need for additional context. We will revise the discussion of Figure 3 to clarify its intent and significance.
>
>
> Visualizations tracking equality loss and cost differences during training: We agree with the reviewer that including plots tracking the evolution of equality loss and relative cost differences throughout training would enhance interpretability. We will include these visualizations in the revised version of the paper to provide deeper insights into the model’s alignment with the OPF objective over time.
>
>
> Formulating OPF for time-varying topologies: We acknowledge the importance of addressing dynamic topologies in real-world power systems and agree with the reviewer that this represents an exciting direction for future research. While our current work focuses on fixed topologies, we plan to extend the framework to accommodate time-varying grid topologies. This will involve evaluating the method on datasets that incorporate real-world operational scenarios, including topology changes and contingencies.

---

### Official Review · Reviewer_JaWa · 2024-11-04

**Soundness:** 1
**Presentation:** 1
**Contribution:** 1
**Rating:** 3
**Confidence:** 4

**Summary:**

The paper presents a GNN-based approach to address the ACOPF problem, with the primary aim of reducing the computation time required by traditional interior-point solvers to enable faster ACOPF solutions. It leverages a physics-informed loss function incorporating penalties for both equality and inequality constraints. The grid model is represented as a graph, having real and reactive power demand, along with node type as inputs. Experimental results are provided, comparing the proposed approach with the MIPS solver across both single and multiple load scenarios.

**Strengths:**

-- Attempting to solve a very relevant problem in the form of ACOPF.
-- Idea of using PINNs and GNN together is a strength.

**Weaknesses:**

— The authors’ claim of being the first to use a PINN for solving the ACOPF problem must be clarified against the various existing works [1]-[2]. There are several works already using PINNs, in both supervised and unsupervised learning settings. There exists a substantial body of work under end-to-end learning that relates to this area, for example: [1]-[2].

— The issue mentioned above seems to stem from an incomplete literature review. For example, [1] presents a survey on end-to-end learning methods for constrained optimization, the general class of optimization problems to which ACOPF belongs. Many works listed there use PINNs in either supervised or unsupervised fashions. Additionally, several key references such as [2]-[4] are not discussed in the paper. Furthermore, the use of GNNs for the ACOPF problem is also not a unique contribution, as noted by the authors themselves. Authors should clearly discuss the limitations of these works and highlight how their work differs from them and provide advantages.

— Experimental studies: The results presented are limited in terms of the number of experiments and comparative analysis. No comparisons are made to existing ML methods.   Authors must compare the results with various ML methods. For example, the cost difference with the proposed method is significantly higher than that achieved by various ML methods for ACOPF.



[1] Kotary, James, Ferdinando Fioretto, Pascal van Hentenryck, and Bryan Wilder. "End-to-End Constrained Optimization Learning: A Survey." In 30th International Joint Conference on Artificial Intelligence, IJCAI 2021, pp. 4475-4482. International Joint Conferences on Artificial Intelligence, 2021.
[2] Seonho Park and Pascal Van Hentenryck. Self-supervised primal-dual learning for constrained optimization. In Proceedings of the AAAI Conference on Artificial Intelligence, volume 37, pp. 4052–4060, 2023.
[3] Ferdinando Fioretto, Terrence WK Mak, and Pascal Van Hentenryck. Predicting ac optimal power flows: Combining deep learning and lagrangian dual methods. In Proceedings of the AAAI conference on artificial intelligence, volume 34, pp. 630–637, 2020.
[4] Priya Donti, David Rolnick, and J Zico Kolter. Dc3: A learning method for optimization with hard constraints. In International Conference on Learning Representations, 2021.

**Questions:**

— If the model is completely unsupervised, why are only 500 demand samples used?

— What are the specifications of the computing hardware cluster (ETH Euler Clusters), and how long are the models trained?

— How does the model shown in Figure 1 differ from existing PINN models? If it does not differ, appropriate citations should be provided.

— In the abstract, the authors highlight the limitation of interior-point methods in achieving global optimality. While this is correct due to the NP-hard nature of ACOPF, how does the current method address this limitation?

--The authors mention that the MIPS equality loss reaches up to 20 MW in case of 118-Bus system Table 2. What does this imply?  Does it mean that MIPS is unable to find a feasible solution? In general, interior point methods yield feasible but potentially suboptimal solutions. Authors should clarify and explain the reasons behind high loss via MIPS, and under what conditions MIPS is failing to converge to any feasible solution.

---

> ### Author Response · Authors · 2024-11-22
> **Reply to reviewer JaWa**
>
> We thank the reviewer for their thoughtful comments and feedback. While we have decided not to proceed further with the submission, we would still like to take this opportunity to clarify and address the points raised.
>
> 1. We appreciate your feedback on this point. To clarify, we do not claim to be the first to use Physics-Informed Neural Networks (PINNs) for solving ACOPF problems. Instead, our contribution lies in successfully achieving zero inequality violation using a PINN in a fully unsupervised framework, which has not been demonstrated in the works you reference.
>
> 2. We recognize the importance of a comprehensive literature review and aim to clearly differentiate our work from existing studies. Below, we provide specific comparisons and highlight the distinctions:
> [1] is a review article that broadly covers end-to-end learning methods for constrained optimization. While it includes methods cited in our work and those mentioned in your comments, our focus is on unsupervised PINN approaches for ACOPF, not general constrained optimization.
> [2] addresses constrained optimization using a Deep Neural Network (DNN) but does not generalize to topological differences in power systems, exhibits inequality violations (unlike our approach), and is tested on only two simple systems with limited configurations (e.g., one generator per node and a single demand).
> [3] was reviewed in our study but not cited because the method is not fully unsupervised, a key focus of our work.
> [4] primarily targets general constrained optimization problems and is tested on a single power system (IEEE57) with 1000 instances, experiencing constraint violations in the test set. This limited scope does not demonstrate the robustness or generality needed for broader ACOPF applications.
>
> 3. You raise an important point about comparisons with existing ML methods. However, our method's unsupervised nature makes direct comparisons with supervised ML approaches less relevant, as those methods typically map traditional ACOPF solutions (obtained via solvers) and do not solve the problem directly. Instead, it will be more relevant to compare our method to traditional optimization solvers, including MIPS and IPOPT, as both solve the problem directly.
>
> 4. Our choice of using only 500 demand samples stems from the principle of avoiding data-intensive methodologies, a primary advantage of unsupervised learning approaches over supervised techniques. However, we acknowledge that increasing the number of samples could provide additional insights. Training with larger datasets would extend training times but does not fundamentally alter the method or its conclusions. The reference [4] reported similarly uses only 1000 instances.
>
> 5. The experiments were conducted on ETH Euler Clusters, which comprise multiple nodes. Detailed specifications can be found on the official Euler webpage. Training times vary depending on the grid size, ranging from several hours to a few days. However, it is important to note that training is a one-time computational cost, after which inference becomes significantly faster than traditional solvers. This advantage is demonstrated in Figure 4.
>
> 6. The model depicted in Figure 1 was custom-developed for this work. To the best of our knowledge, no other studies use the same architecture combined with our PINN loss function.
>
> 7. You correctly note that achieving global optimality is a limitation of nonlinear optimization problems, including ACOPF. Our method does not claim to resolve this inherent challenge. Instead, we aim to provide an alternative to traditional nonlinear solvers for ACOPF problems, demonstrating competitive performance and practical advantages, such as zero inequality violations.
>
> 8. The equality loss of up to 20 MW observed for the 118-bus system does not imply that MIPS fails to converge. Rather, it reflects the fact that equality constraints are treated as hard constraints and minor violations can occur within the solver's tolerance. Importantly, MIPS would not converge if inequality constraints were violated beyond permissible limits or if an initial condition was infeasible.

---

> > ### Comment · Reviewer_JaWa · 2024-11-25
> > **Response to Rebuttal**
> >
> > I appreciate the authors' point-by-point response. However, I have the following reservations regarding the motivation and focus:
> >
> > 1. In existing unsupervised or supervised learning methods, inequality constraint satisfaction has not been identified as a major limitation. For example, the supervised method DC3 achieves zero inequality gaps, and unsupervised methods, such as Park & Pascal (2023), report maximum inequality gaps of $10^{-3}$ for the 118-bus system. What are the numerical values of inequality gaps in the proposed method? Furthermore, do the authors have any theoretical guarantees on achieving zero inequality gaps? This clarification is necessary.
> >
> > 2. The authors state in their rebuttal that training time ranges from **several hours to a few days**, depending on the system. The authors must clarify: **Why would one train an unsupervised model for days when similar performance could be achieved using some supervised data, and smaller total time?** For instance, generating 1,000 data points and training a DC3-type model would be much cheaper than training for several days. Additionally, it would be important to understand how the training times of the proposed method scale with system size.
> >
> > 3. I respectfully disagree with the authors' point that comparisons with supervised models are not useful. State-of-the-art (SOTA) supervised models achieve similar levels of performance with much less total time (i.e., data generation time + training time). Thus, they must be included as benchmarks for a fair comparison.
> >
> > 4. The authors' point regarding MIPS convergence is unclear. A 20 MW equality gap is not a minor violation within the solver's tolerance. Even when expressed in per-unit terms (on a 100 MVA base), a 0.2 per-unit equality gap is substantial. By contrast, most methods (both supervised and unsupervised) listed in Table 3 of Park & Pascal (2023) achieve equality gaps that are one order of magnitude smaller than 0.2. For optimization solvers, equality gaps should ideally be less than 0.0001 to ensure convergence.
> >
> > 5. How does proposed method generalize over grid topologies: theoretically and empirically?
> >
> > 6.  When unlabeled data is inexpensive (e.g., simple load sampling), limiting the approach to 500 samples to avoid data-intensive methodologies seems counterintuitive. The primary limitation of supervised models arises when generating large amounts of labeled data by solving ACOPF instances, which is computationally expensive. However, when only unlabeled data (e.g., load points) is needed, why restrict the dataset to just 500 samples? The example cited from reference [4], which uses 1,000 data points, appears misplaced in this context because those are labeled data points generated by solving ACOPF, not sampled load points.

---

> > ### Author Response · Authors · 2024-11-26
> > **Reply to jaWa**
> >
> > We thank the reviewer for their reply and for raising this important question.
> >
> > 1.	Regarding inequality constraint satisfaction, we acknowledge that supervised methods like DC3 and some unsupervised methods (e.g., Park & Pascal, 2023) report negligible inequality gaps for test systems. Both references cited provide results for limited cases and report their findings in brief, without delving into theoretical guarantees of achieving zero inequality gaps. To address the reviewer’s point, we emphasize that the DC3 paper does not explicitly provide a theoretical guarantee for zero inequality violations. Instead, its results are empirically validated. Similarly, our approach experimentally achieves zero inequality violations (~10−6) across the test systems considered, comparable to DC3. If the reviewer could point us to a section in DC3 or related works where such a theoretical guarantee is described, we would be glad to explore it further and potentially adapt similar reasoning to our framework. Lastly, we note that while experimentally achieving zero inequality violations is a valuable outcome, the challenges of extending this to a broader range of systems and ensuring robustness under diverse conditions remain areas for future investigation in this field.
> >
> > 2.	Regarding the training times of DC3 and Park & Pascal, we kindly ask if the reviewer could point us to where the training time and dataset creation time are explicitly reported in these methods. From our review of the DC3 paper, only inference times are reported, and Park & Pascal state that solving (training) the instance takes approximately 120 minutes. While we acknowledge the longer training times, this tradeoff is an expected aspect of developing an unsupervised framework. To address concerns about scalability and training time reductions, we note that using GPUs instead of CPUs (as in our experiments) reduces training time by approximately fivefold, depending on the system. Further, while the training time scales with the system size, the inference time of our model remains constant, enabling rapid generalization to unseen conditions. Reducing training time may be critical for real-world deployment, and exploring hybrid supervised-unsupervised approaches or other efficiency improvements could be an avenue for future work, though it is not the focus of this paper. Finally, we appreciate the reviewer’s suggestion to analyze how training time scales with system size, we agree that it is a relevant question to explore.
> >
> > 3.	We respectfully maintain that comparing our unsupervised method to supervised models is not the most appropriate benchmark, as the two approaches have fundamentally different objectives. Supervised models, such as DC3, aim to replicate the solutions of a solver, inheriting the solver’s tendencies and local minima. Our unsupervised method, by contrast, directly solves the AC-OPF problem without relying on pre-computed solutions, potentially identifying different valid local minima. Why should we compare our unsupervised method to supervised models that aim to replicate the solution of a solver, when we can directly compare our method to the solver itself?
> >
> > 4.	We acknowledge that the equality gaps reported in our current submission, when using MIPS as the baseline solver, may appear larger than what is expected for optimization solvers. To address this concern, we will recompute these results using IPOPT, as employed in Park & Pascal (2023).
> >
> > 5.	Currently, our work primarily establishes the potential of the method without presenting explicit empirical evidence on varying grid topologies. However, we recognize the necessity of demonstrating this capability, both theoretically and empirically. We plan to include additional experiments in future iterations of our work. These experiments will test the method's performance on grids with varying topologies. These results will provide empirical evidence to support our claim about the method's generalization capabilities. Theoretically, the use of a graph neural network (GNN) as part of the architecture inherently allows for flexibility across different grid topologies. The GNN leverages the graph structure of the grid, making it topology-agnostic in terms of its operations.
> >
> > 6.	The choice to limit the dataset to 500 samples was deliberate and aimed at highlighting the generalization capabilities of our approach. Our method requires a more computationally intensive process in training rather than the input data generation (which is rather trivial and inexpensive, as rightfully pointed out). Our focus was to demonstrate that the method could generalize even with a relatively small training dataset. That said, we agree that increasing the dataset size could improve the performance and provide additional insights. This could be useful for larger and more complex systems. We will explore this aspect in future work to assess the trade-off between dataset size, training time, and performance.

---

> > > ### Comment · Reviewer_JaWa · 2024-11-27
> > > **Reply to Authors**
> > >
> > > It is interesting that the authors argue that benchmarking directly with an optimization problem solver is more appropriate than comparing with other ML methods. I would like to raise a few points for clarification and discussion:
> > >
> > > 1. **Theoretical Guarantee of Feasibility**:
> > >    The authors claim that their method **directly solves the AC-OPF problem without relying on pre-computed solutions**, distinguishing it from supervised approaches that mimic a solver. If this is the case, I strongly believe that the method should provide **theoretical guarantees** on the feasibility of the AC-OPF problem concerning both equality and inequality constraints. A key advantage of solvers is their ability to provide feasible solutions consistently. If the claim is to compete directly with solvers, then a guarantee of feasibility must be explicitly demonstrated.
> > >
> > > 2. **Feasibility Gap in MIPS Results**:
> > >    The authors acknowledge that the MIPS results exhibit unusually high feasibility gaps. This raises concerns about the interpretation of the results. Specifically, I find a 20 MW gap in equality constraints to be excessively large. For comparison, consider an approximation of the AC-OPF problem, such as polynomial regression, which achieves a gap of 16 MW. Should this be considered superior to MIPS or other solvers under the authors’ framework? The authors need to clarify how such discrepancies are addressed in their analysis.
> > >
> > > 3. **Comparison with Self-Supervised Methods**:
> > >    While the authors argue that comparing with supervised methods is unfair, they should still present a comparison with self-supervised approaches like Park & Pascal. Additionally, Park & Pascal provide a detailed table on data generation times, and their DC3 code is publicly available. I suggest the authors include a runtime comparison using the same computational setup to provide a fair benchmark for evaluating results.
> > >
> > > 4. **Test Case Size and Scope**:
> > >    The authors’ point about limited test cases in prior work requires further scrutiny. Both this paper and Park & Pascal evaluate up to 118-bus systems. However, numerous works in the power systems literature test much larger systems. For example, works like **DeepOPF** and others listed in [this wiki on ML-OPF](https://energy.hosting.acm.org/wiki/index.php/ML_OPF_wiki) evaluate significantly larger systems. The authors should clarify how their method scales to larger test cases compared to these existing works.

---

### Meta-Review · Area_Chair_wh9D · 2024-12-20

**Metareview:**

This paper addresses the challenges of integrating intermittent renewable energy sources, by enhancing the Alternating Current Optimal Power Flow (AC-OPF) process. AC-OPF is a fundamental optimization problem in power systems that ensures safe and cost-effective grid operation. While traditionally solved using linearized approximations or non-linear solvers like the interior point method, these approaches face limitations, such as inaccuracies, convergence issues for large systems, and lack of global optimality.

The paper introduces PINCO, a physics-informed graph neural network (GNN) framework designed to solve AC-OPF efficiently. PINCO models the power grid as a graph, incorporating real and reactive power demand along with node types as inputs. It employs a physics-informed loss function that enforces equality and inequality constraints, enabling the method to produce accurate and feasible solutions. Unlike traditional methods, PINCO achieves results in a fraction of the computational time, generalizes effectively across diverse loading conditions, and handles multiple generators per bus with minimal adjustments to hyperparameters.

Experimental results demonstrate that PINCO outperforms traditional solvers, such as the MIPS solver, in both single and multiple load scenarios. It functions as both a solver and a hybrid universal function approximator, ensuring compliance with inequality constraints while maintaining adaptability to various power systems. Overall, PINCO addresses the critical need for scalable, fast, and accurate solutions to AC-OPF, paving the way for better management of power grids in the context of the energy transition. It highlights the potential of machine learning, particularly GNNs, in optimizing complex, non-linear power system challenges.

The reviewers raise several major issues, the most important of which is the lack of a thorough and fair review of the existing literature. This leaves the paper's contribution unclear, given the existing literature. Furthermore, the reviewers criticize the novelty, the level of empirical evidence, and the actual improvements brought about by the proposed PINCO approach. Collectively, the concerns significantly outweigh the merits of the paper.

The authors are encouraged to consider the reviewers' comments to improve the contribution and presentation of the paper.

**Additional Comments On Reviewer Discussion:**

The reviewers have pointed out issues about the relevance to the existing literature, the contribution, interpretation of the observations, and the adequacy of the empirical studies. The authors, for the most part, have acknowledged these issues. While I can believe that the authors can revise the manuscript to address some of the issues, I believe the level of revision required exceeds what is customary and needs a thorough overhaul. This is in addition to the concerns about the novelty of the approach.

---

### Decision · Program_Chairs · 2025-01-22

Reject